# Serological Tests in the Detection of SARS-CoV-2 Antibodies

**DOI:** 10.3390/diagnostics11040678

**Published:** 2021-04-09

**Authors:** Kissy Guevara-Hoyer, Jesús Fuentes-Antrás, Eduardo De la Fuente-Muñoz, Antonia Rodríguez de la Peña, Marcos Viñuela, Noemí Cabello-Clotet, Vicente Estrada, Esther Culebras, Alberto Delgado-Iribarren, Mercedes Martínez-Novillo, Maria José Torrejón, Rebeca Pérez de Diego, Miguel Fernández-Arquero, Alberto Ocaña, Pedro Pérez-Segura, Silvia Sánchez-Ramón

**Affiliations:** 1Department of Immunology, IML and IdISSC, Hospital Clínico San Carlos, 28040 Madrid, Spain; kissgh@gmail.com (K.G.-H.); eduardodelafuentemunoz@gmail.com (E.D.l.F.-M.); Arpena@salud.madrid.org (A.R.d.l.P.); vinuelamarcos@gmail.com (M.V.); mfarquero@salud.madrid.org (M.F.-A.); 2Department of Immunology, Ophthalmology and ENT, School of Medicine, Complutense University, 28040 Madrid, Spain; 3Department of Medical Oncology, Hospital Clínico San Carlos, 28040 Madrid, Spain; jfuentesantras@outlook.com (J.F.-A.); alberto.ocana@salud.madrid.org (A.O.); pedro.perez@salud.madrid.org (P.P.-S.); 4Unit of Infectious Diseases, Department of Internal Medicine, Hospital Clínico San Carlos, 28040 Madrid, Spain; ncabello@salud.madrid.org (N.C.-C.); vicente.estrada@salud.madrid.org (V.E.); 5Department of Microbiology, IML and IdISSC, Hospital Clínico San Carlos, 28040 Madrid, Spain; esther.culebras@salud.madrid.org (E.C.); adelgadoi@salud.madrid.org (A.D.-I.); 6Clinical Analysis Department, IML and IdISSC, Hospital Clínico San Carlos, 28040 Madrid, Spain; mercedes.martineznovillo@salud.madrid.org; 7Department of Biochemistry, IML and IdISSC, Hospital Clínico San Carlos, 28040 Madrid, Spain; mariajosefa.torrejon@salud.madrid.org; 8Laboratory of Immunogenetics of Human Diseases, IdiPAZ Institute for Health Research, 28029 Madrid, Spain; rebeca.perez@idipaz.es

**Keywords:** SARS-CoV-2, antibodies, serological tests, healthcare personnel

## Abstract

Early detection of SARS-CoV-2 is essential for a timely update of health policies and allocation of resources. Particularly, serological testing may allow individuals with low-risk of being contagious of SARS-CoV-2 to return to daily activities. Both private and academic initiatives have sought to develop serological assays to detect anti-SARS-CoV-2 antibodies. Herein, we compared five different assays in active healthcare personnel exposed to SARS-CoV-2 in a large center in Madrid, Spain, in a retrospective study. Median time lapse between polymerase chain-reaction (PCR) and serological testing was 11 days (7–21). All tests assessed IgM/IgG titers except for Euroimmun (IgA/IgG) and The Binding-Site (IgA/IgM/IgG). The highest concordance rate was observed between Dia.Pro and Euroimmun (75.76%), while it was lowest between The Binding-Site and Euroimmun (44.55%). The Binding-Site assay showed the highest concordance (85.52%) with PCR results. Considering PCR results as reference, Dia.Pro was the most sensitive test, although The Binding-Site assay exhibited the highest area under the curve (AUC; 0.85). OrientGene and MAGLUMI tests were performed in a smaller cohort with confirmed infection and thus were not adequate to estimate sensitivity and specificity. The Binding-Site assay presented the best joint sensitivity and specificity among all the tests analyzed in our cohort. Likewise, this serological assay presents a greater repertoire of antibodies and antigen-regions tested, which is why each individual’s humoral immunity is more accurately reflected. The better the immunity test, the most adequate the health strategy to take in terms of organization of consultations, surgery, and treatments in vulnerable patients. The three antibody classes (IgG/IgM/IgA) were determined jointly, which translates to an economic impact on healthcare. While their role in the protection status remains elusive, serological tests add a valuable tool in the early management of SARS-CoV-2 after known exposition.

## 1. Introduction

The COVID-19 pandemic represents a global health concern with unprecedented social and economic repercussions. SARS-CoV-2 virus is a positive-strand RNA virus with a genome that encodes structural proteins comprising the spike (S), envelope (E), membrane (M), and nucleocapsid (N) [1,2]. The incubation period may vary among different patient cohorts [2,3]. Viral RNA can be detected in swab samples until approximately day 14 post infection [4,5]. IgM SARS-CoV-2 specific antibodies have been detected from day 3 post illness onset and/or initial exposure [2] in asymptomatic patients (Figure 1). IgM reaches a maximum peak between weeks 2 and 3, and it can be detected up to 1 month following exposure to the virus [6,7,8]. Both IgA and IgG SARS-CoV-2 specific antibodies are detected from day 4 post illness onset, increasing gradually until reaching a peak after 2 weeks. However, the kinetics of IgA and IgG curves tend to be less accentuated than that of IgM [6,7,9]. In the late phase, the detection of the specific antibodies becomes more variable, with heterogenous reported evidence on their levels [6,10,11].

Serological tests that identify past infection can be used to estimate cumulative incidence, but the relative accuracy and robustness of various sampling strategies has been unclear [12]. Early detection of SARS-CoV-2 is a crucial intervention to control virus spread and dissemination [2]. Polymerase chain-reaction (PCR)-based tests provide information on the presence of the virus in the nasopharyngeal cavities and therefore identify potential contagious carriers either symptomatic or asymptomatic [13]. In addition to PCR tests, immune-based assays that determine exposure to SARS-CoV-2 and protection status are required to enable better-informed decisions, especially whether low-risk individuals should return to the daily activities [12].

The different tests for the determination of SARS-CoV-2 infection are more reliable when there is a compatible medical history, examination, and imaging test [14]. However, in the case of asymptomatic individuals or regions where the disease prevalence is low, these results must be carefully analyzed. It is vitally important to know the advantages, limitations and usefulness of each specific test to determine its diagnostic accuracy and use accessory diagnostic tests that allow the results to be thoroughly analyzed, always taking into account their individual and societal implications.

In general, the sensitivity of antibody tests is too low in the first week since SARS-CoV-2 exposure and has only been extensively evaluated in patients that required hospitalization [15,16]. Therefore, they cannot be used as a SARS-CoV-2 diagnostic tool in isolation, being able to present lower levels of antibodies in the population with mild or asymptomatic symptoms. Nevertheless, the serological test offers an essential complementary role in individuals with a high risk of contagion due to sustained exposure when RT-PCR tests are negative or are not done [16].

To date, many commercial companies and research institutes have developed and compared different serological assays to detect anti-SARS-CoV-2 antibodies from patient serum or plasma samples [2,9,10,16,17,18,19,20]. These assays mainly target immunogenic coronavirus S protein, which is the most abundant viral protein and where the receptor-binding domain is located, and N protein, which is profusely expressed during infection [21,22,23]. In this study, we analyzed the concordance rates among five different serological tests and with PCR results to better characterize their utility in the early detection of SARS-CoV-2 infection.

## 2. Methods

We conducted a single-center, retrospective cohort study to describe the diagnostic potential of five serological assays in personnel exposed to SARS-CoV-2 in a large tertiary hospital in Madrid, Spain. We collected 258 samples derived from a random screening of a cohort of active health workers (doctors, nurses and assistant nurses mainly) in direct and constant exposure to patients with SARS-CoV2 infection during the maximum incidence of infection recorded in the hospital during the pandemics. The rationale of the study was the urgent need to validate the serological status in health staff who were going to take care of vulnerable patients (cancer patients and immunodeficiency patients) in the first place. The samples collection was motivated by the need for ongoing clinical validation studies for internal regulatory approval of SARS-CoV-2 diagnostic tests in our center.

A total of four of the tests were ELISA-based assays and the remaining test was performed by lateral flow assay. Both antibody detection methods were proposed as valid tools for detecting antibodies against SARS-CoV-2 in the Report for analytical techniques in COVID-19 presented by the Spanish Society of Immunology (Version 01. 30 November 2020) [24].

The 258 samples were drawn and collected from a time interval between March 30 to 3 April 2020 (period of maximum incidence of infection recorded in the hospital). PCR was performed in 224 individuals as part of the preventive measures to control the exposure of risk personnel or as routine practice in symptomatic individuals (negative: 138 samples, positive: 86 samples). Serological and PCR results were further contrasted in each participant (Figure 2). PCR performance period varied between the 7th and the 21st day prior to obtaining the serum sample for serological study. The study was conducted according to the guidelines of the Declaration of Helsinki, and approved by the Institutional Review Board. The study was approved by the Ethics Committee of the Hospital Clinico San Carlos (20/243-E_BS). Written informed consent was waived given the emergency of the current pandemic.

### 2.1. ELISAs

The four SARS-CoV-2 ELISA-based serological assays evaluated included: human IgG/IgM anti-SARS-CoV-2 ELISA by MAGLUMI (SNIBE—Shenzhen New Industries Biomedical Engineering Co., Ltd., Shenzhen, China); human IgG/IgA anti-SARS-CoV-2 ELISA by EUROIMMUN (Medizinische Labordiagnostika AG, Lubeck, Germany); enzyme-immuno-assay for IgG/IgM antibodies to COVID-19 by Dia.Pro (Diagnostic Bioprobes S.r.l., Milan, Italy); and Human IgG/IgA/IgM anti-SARS-CoV-2 ELISA by The Binding Site Group Ltd., Birmingham, UK (Table 1). Serum was extracted within 4 h from blood draw and samples were stored at −80 °C and thawed before the analysis. The SARS-CoV-2 receptor binding domain (RBD) of the S protein was the target to determine the antibody titers. Two of the four tests (MAGLUMI and Dia.Pro) also detected antibodies against the N protein from the coronavirus nucleocapsid. In ELISA-based SARS-CoV-2 serologies, the optical density (OD) was assessed at 450 and 620 nm on a plate reader and was adjusted subtracting both OD. To estimate antibody titers, we generated isotype-specific standard curves using anti-SARS-CoV-2 monoclonal measure IgG, IgA, and/or IgM antibodies and implemented it to calculate the concentration of anti-RBD of each antibody. Positive specimens were identified as those with an UR/mL three standard deviations above the mean negative control specimens, following the manufacturer’s instructions. We adjusted the estimates of prevalence for the sensitivity and specificity for each antibody isotype and each test, as well as the detection rate of any isotype.

### 2.2. Lateral Flow Assay (LFA)

The Orient Gene Biotech COVID-19 IgG/IgM Rapid Test Cassette was performed by lateral flow assay (Zhejiang Orient Gene Biotech Co., Ltd., Anji County, Huzhou, China). Orient Gene Biotech Rapid Test Cassette and MAGLUMI tests were performed only on samples associated with confirmed positive PCR.

### 2.3. Statistical Analysis

Concordance rates were calculated among the five serological tests and between each test and matched PCR results when available. Two SARS-CoV-2 serologies (MAGLUMI and The Orient Gene Biotech) were performed exclusively in forty-five patients with known positive PCR for SARS-CoV-2. A receiver operating characteristic (ROC) curve was calculated for each test illustrating their diagnostic ability by plotting the true positive rate (sensitivity) against the false positive rate (1-specificity) at various threshold settings. For ROC analysis, PCR results were considered as reference to estimate sensitivity and specificity. MAGLUMI and The Orient Gene Biotech assays were excluded from ROC analysis since they were performed only in PCR-positive patients. Statistical Product and Service Solutions (SPSS) software version 20 (Chicago, IL, USA) was used for descriptive and statistical data analysis. Contingency analysis was performed using GraphPad Prism version 8.3.0 for Windows (GraphPad Software, La Jolla, CA, USA). The area under the Receiver Operating Characteristic curve (ROC) of the model was calculated with the ROC method implemented in the SPSS statistics software. *p* < 0.05 was considered statistically significant.

## 3. Results

We evaluated specific antibodies against SARS-CoV-2 using ELISA or lateral flow assay in 258 serum samples from individual active health workers. The data of concordance among serological tests are displayed in Table 2, being highest for Dia.Pro and OrientGene (82.22%) and lowest for Euroimmun and The Binding Site (44–55%). Since PCR tests are the most widespread tool to detect and follow COVID-19 infection, we compared the results of the serological assays with matched PCR tests. Importantly, no significant differences were found among the group of patients analyzed through each serological assay regarding the time lapse between PCR and blood sampling (*p* = 0.4). The concordance rate between PCR results and serological tests is shown in Figure 3, being highest for The Binding Site (85.52%) and lowest for MAGLUMI (68.89%).

To further characterize the diagnostic capability of the five serological assays, we performed a receiver operating characteristic (ROC) curve analysis using PCR as reference (Figure 4). Of note, MAGLUMI and Orient Gene assays were performed only in PCR-positive individuals and therefore, sensitivity and specificity could not be estimated. The Binding Site and Euroimmun assays showed good overall performance (AUC 0.85 and 0.84, respectively), slightly better for the first assay in which also a bigger population was available for analysis (221 vs. 191, respectively). The Dia.Pro test exhibited the highest sensitivity (0.93) but a poor specificity (0.57).

## 4. Discussion

SARS-CoV-2 serological tests are designed to detect antibodies against specific viral proteins [1,2,6,17,21]. The S protein is responsible for the virus binding to the host, its tropism, and its capacity for transmission [17,18,25,26]. The N protein is a structural component that plays an essential role in the pathogenesis and viral replication, especially at early phases of infection [17,26,27]. Currently, both proteins provide the most immunogenic viral antigens [21,22,23]. Burbelo et al. reported that the nucleocapsid protein is more sensitive to specific anti-SAR-CoV-2 antibodies detection than spike protein in the early phase of the infection [26]. Therefore, a test that determines both antigenic regions will have a greater capacity to detect antibodies against infection at different disease stages, which is especially relevant if the asymptomatic population is considered. SARS-CoV-2 serological tests aimed at quantifying more than one of these proteins simultaneously could achieve greater sensitivity and specificity.

The diagnostic capacity of SARS-CoV-2 serological tests increases from 10–20 days after infection [6,7,9]. Previous studies showed in a meta-analysis study that antibody tests have higher sensitivity from 15 days after symptom onset and 99% specificity for detecting SARS-CoV-2 [15,16]. Likewise, the test’s sensitivity may be affected depending on the time of the disease when the test is performed. Carpenter et al. showed significant differences in the sensitivity of the test, depending on whether it is a molecular test or a serological antibody test (95% vs. 56%) [14], which may be due, among other reasons, to the characteristics of the population studied as well as derived from the performance of the test in early stages of the disease (<15 days).

However, different factors can influence the proper interpretation of the results and, therefore, standardization of a specific technique [28]. The main factors involved are: the isotype of antibodies detected (IgG/IgM and/or IgA), the antigenic region of the RBD protein to which the test is directed, the time-point of the illness when the test is performed, the immune profile of each individual, and the method by which it is performed (ELISA, lateral flow assay, etc.).

Regarding the advantage of using a specific technique, commercial ELISA assays and LFA tests can be used as complementary tools in COVID-19 diagnosis. Nevertheless, previous studies found a variable performance of the different LFA tests [29,30,31]. For this reason, LFA tests are not considered as an adequate single strategy for SARS-CoV2 immunity diagnosis. As a general principle, the assays predictive values depend on the SARS-CoV-2 geographic prevalence, and our results are proposed as a diagnostic tool in the setting of our specific population during the pandemics. Serological results must be interpreted in conjunction with clinical symptoms, PCR tests, and additional laboratory profiling [13]. From day 14 on, the viral load tends to decrease and fall below the detection threshold of PCR tests, and consequently, real-time comparison between the two tests is constrained to a short time frame [4,5].

It is necessary to emphasize that the PCR shows several drawbacks, being a time-consuming technique for sample ribonucleic acid extraction, PCR reaction of approximately 6 h, thus 12–24 h use to be necessary to get the result. Providing fast results is part of an efficient strategy for COVID-19 diagnosis in the pandemics context [15]. Thus, researchers such as Byrnes et al. propose quicker methods, such as the multiplexed amplification of SARS-CoV-2 RNA from nasopharyngeal swabs. These strategies could reduce complexity, time, and costs for detecting COVID-19, maintaining high sensitivity and specificity (86% sensitivity and 100% specificity) [32]. However, these methods require validation in different population cohorts.

A significant proportion of the population may remain asymptomatic during the infection, thus limiting the usefulness of targeted PCR in high-risk communities [17,33]. In the case of healthcare personnel, asymptomatic carriers represent a critical vulnerability in the protection of patients and also imply a greater risk of contagion in the working environment, thus affecting the management of human resources. Anti-SARS-CoV-2 serology is currently considered a diagnostic aid [10,11]; nevertheless, SARS-CoV-2 antibodies could enable a quantitative determination of disease prevalence, especially in high-risk communities.

A moderate-to-high concordance was observed among the five serological assays evaluated, and concordance with PCR results was found to be over 80% for The Binding Site and Euroimmun assays. PCR tests are considered the reference tool for diagnosis and follow-up in daily care. Therefore, we performed a ROC analysis showing that The Binding Site and Euroimmun tests provided the best fit for sensitivity and specificity of COVID-19. These tests were positive in 82% and 80% of individuals with positive PCR, respectively, at the cost of an acceptable 12% false positive rate (FPR or 1-Specificity), well under the 43% FPR that may hinder the implementation of the Dia.Pro assay in the clinical setting. We may acknowledge that these data illustrate only the internal validity of the tests, but do not tackle their external validity in the multiple and varying epidemiological contexts in which disease prevalence and predictive values may not be accurately estimated. In this regard, while the detection of potential asymptomatic carriers is critical to prevent the spread of the virus, also a specificity threshold must be sought to avoid overdiagnosis and secure an appropriate allocation of resources.

Discordant results between PCR and serological tests could be explained by different reasons, including improper sampling, timing during infection, or specimen handling, among others [1,2,13]. Similarly, autoimmune diseases, high levels of rheumatic factor, several viral infections as cytomegalovirus, Epstein-Barr Virus, HIV, hepatitis B virus, and toxoplasma infection could also confuse results [13,21,24]. Importantly, all of these analyses were performed in a population of active healthcare workers with known exposure to COVID-19, and with a median time between PCR and blood sampling of 11 days (7–21), which was not different when compared across the different serological tests (*p* = 0.4). These considerations may reinforce the appreciation that serological tests, and particularly The Binding-Site and Euroimmun, can robustly detect SARS-CoV-2 at early phases of infection.

This study must be considered exploratory and has some limitations, including the lack of clinical data from individuals and the difficulty to assess the protection status given that serological tests were performed only once and not repeated. Overall, serological tests assessing IgG and IgM or IgA may convey timely diagnostic information and inform clinical decisions in early phases of SARS-CoV-2 infection.

The Binding-Site assay presented the best joint sensitivity and specificity among all the tests analyzed in our cohort. Likewise, this serological assay presents a greater repertoire of antibodies and antigen-regions tested, which is why each individual’s humoral immunity is more accurately reflected. The better the immunity test, the most adequate the health strategy to take in terms of organization of consultations, surgery, treatments in vulnerable patients. The three antibody classes (IgG/IgM/IgA) were determined jointly, which translates to an economic impact on healthcare.

## Figures and Tables

**Figure 1 diagnostics-11-00678-f001:**
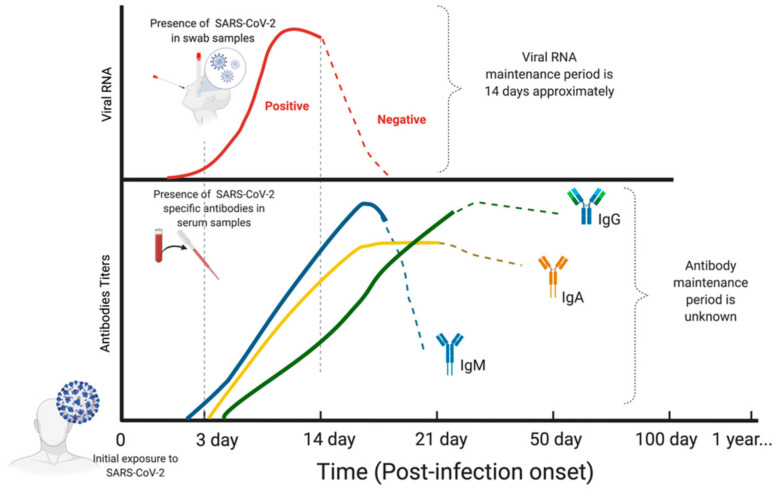
Detection of SARS-CoV-2 by RNA testing and specific antibodies (IgG, IgM, IgA) plotted according to the time since the initial exposure to the virus. Dashed lines denote the variable dynamics of adequate levels of SARS-CoV-2 specific antibodies. Figure created with www.BioRender.com (accessed on 8 April 2021).

**Figure 2 diagnostics-11-00678-f002:**
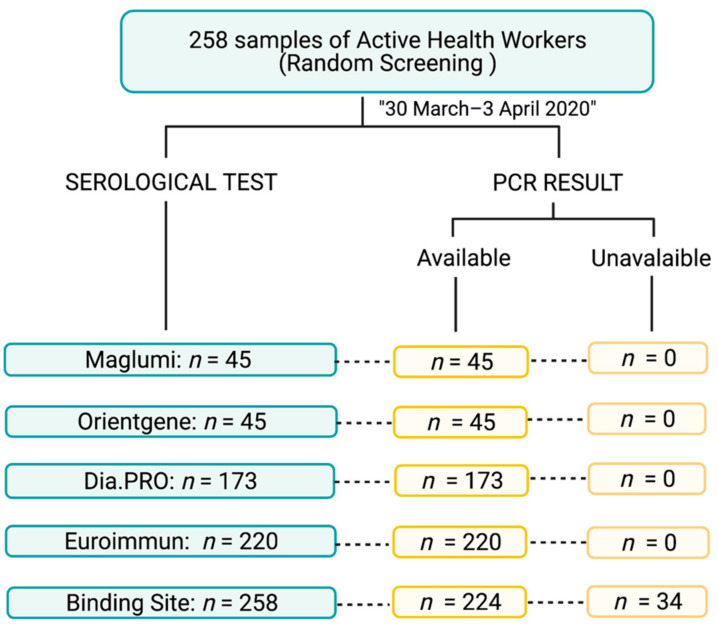
Flowchart indicating the number of samples analyzed with each serological test and the availability of an associated PCR result. The samples were analyzed with each serological test as they arrived at the laboratory and using all the tests available at our center at the study time. Created with www.BioRender.com (accessed on 8 April 2021).

**Figure 3 diagnostics-11-00678-f003:**
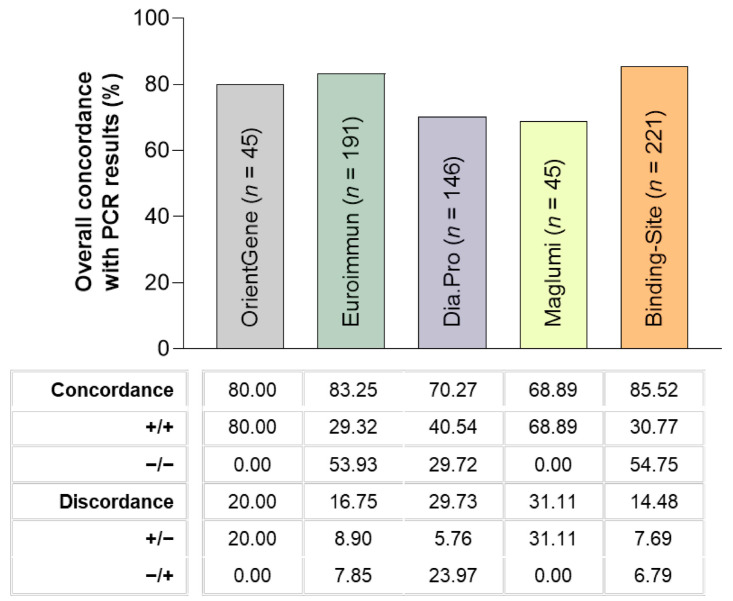
Concordance between each SARS-CoV-2 serological test and matched PCR results.

**Figure 4 diagnostics-11-00678-f004:**
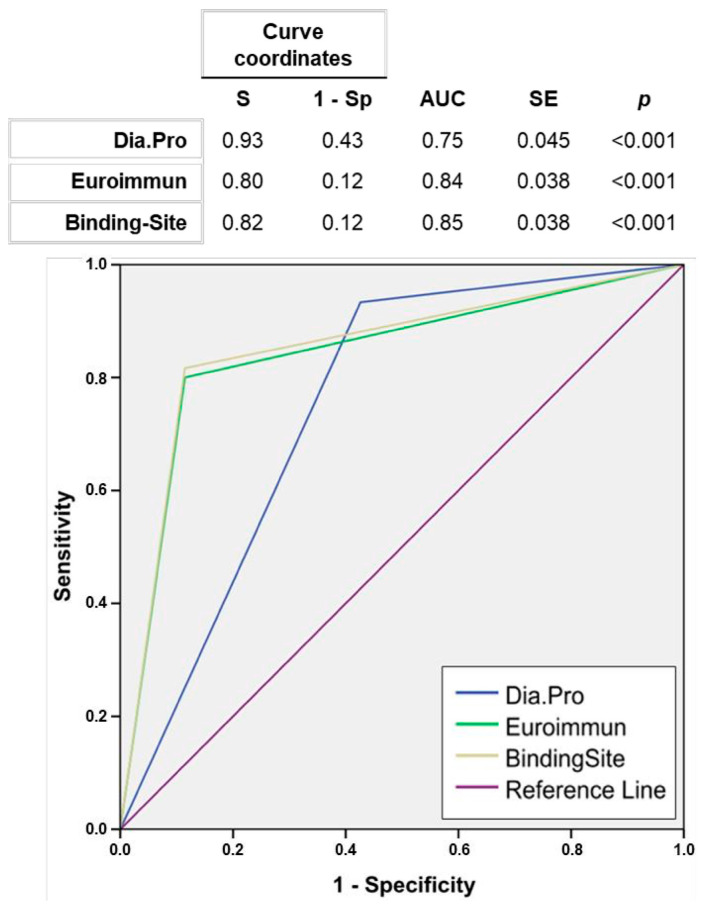
Receiver operating characteristic (ROC) curves of serological assays. Abbreviation: S, sensitivity. Sp, specificity. AUC, area under the curve. SE, standard error.

**Table 1 diagnostics-11-00678-t001:** Main characteristics of each serological test evaluated.

Test	Anti-SARS-CoV-2 Antibodies Isotype	Serological Assays	Antigenic Region of the RBD Protein
MAGLUMI	IgG/IgM	ELISA	N and S
EUROIMMUN	IgG/IgA	ELISA	S (S1)
Dia.Pro	IgG/IgM	ELISA	N and S
The Binding Site Group Ltd.	IgG/IgA/IgM	ELISA	S (Trimer)
The Orient Gene Biotech	IgG/IgM	LFA	N

Abbreviation: LFA, Lateral Flow Assay. N, Nucleocapsid. RBD, receptor-binding domain. S, Spike.

**Table 2 diagnostics-11-00678-t002:** Concordance among five different SARS-CoV-2 serological tests.

Tests					Samples	Positive (%)	Negative (%)	Total (%)
					Concordance			
Dia.Pro	Binding-Site	Euroimmun	Maglumi	OrientGene	45	55.55	4.44	60.00
Dia.Pro	Binding-Site	Euroimmun			132	38.63	33.33	71.97
Dia.Pro	Binding-Site				171	36.84	38.59	75.44
Dia.Pro		Euroimmun			132	42.42	33.33	75.76
	Binding-Site	Euroimmun			220	29.54	15.00	44.55
Dia.Pro			Maglumi		45	66.66	4.44	71.11
Dia.Pro				OrientGene	45	77.77	4.44	82.22

## Data Availability

The data supporting the findings of this study are available within the article. Raw data compliant with the institutional and home country confidentiality policies can be available upon request from the corresponding author.

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
