# Peer review of "Serological Tests in the Detection of SARS-CoV-2 Antibodies"

_diagnostics, 2021, doi:10.3390/diagnostics11040678_

Round 1
Reviewer 1 Report
The manuscript entitled Serological Tests in the Detection of SARS-CoV-2 Antibodies evaluates the sensitivity and specificity of several commercial serological tests and their correlation with PCR based test.
The concept of the study is good; however, the study warrants significant revision before consideration for further review. Please see the comments below.
- The description of the patient cohort should be stated. It is not very clear the importance of the term “individual active health workers”. Are the individual samples from healthy donors or recovered patients? How relevant is it to study the antibody titer in patients with unknown element of SARS-CoV-2 exposure? Authors should explicitly explain the rationale of using particular patient population. Or this is a random screening of the population for the SARS-CoV-2 infection.
- The legend of Y axis of the Figure 1 should be stated.
- It is also unclear how authors draw conclusions on the performance of each diagnostic test in Figure 3 based upon the outcome observed in different assays without providing further explanation. What does each color curve represent? This is not evident in neither the text nor in the Figure legend?
- The authors describe the antibody isotypes recognized by the 5 different detection tests that are being studied, however they are not stating the principles of these assays in terms of region/antigenic region of the RBD protein that have been utilized to develop the product. This analysis will help us to understand why one type of assay is more sensitive and specific compared to the other.
Minor comment
The term ROC is first being used in statistical analysis section needs to be stated. It is described later in the result section, it will be helpful to discuss the significance of this type of analysis early on.
Reviewer 2 Report
This is a single-center, retrospective cohort study describing the diagnostic potential of 5 serological assays in personnel exposed to SARS-CoV-2. The study was performed in a large tertiary hospital in Spain. Overall, the study is well done and the concept is clear. The paper is unmet need. However there are several critical points that need attention:
Abstract: please precise that the study was retrospective. Please consider inserting a graphical abstract that could explain the study and its aims (this will rise visibility and appeal).
Methods section: were there inclusion and exclusion criteria? Please precise. Question: can outcomes of ELISA testing and lateral flow assay be compared? Please explain in the methods section.
Section 2.1 ELISAs: please consider creating a table on the data reported by synthetizing the testings. This is for better understanding.
Results: Please indicate at the beginning the overall number of positive between the 258 serum samples. Please create a flowchart indicating how testings were done and the number tests per each. As it is done now it is difficult to understand. it is not clear if same thestings were done on each positive patient. This is essential to know in order to better compare the tests.
Discussion: please discuss and cite Multiplexed and Extraction-Free Amplification for Simplified SARS-CoV-2 RT-PCR Tests (PMID: PMID: 33631932); Antibody tests for identification of current and past infection with SARS-CoV-2 (PMID: 32584464). Please also discuss sensitivity of testings. See and cite: PMID: 33197346, and Rapid antigen and molecular tests had varied sensitivity and ≥97% specificity for detecting SARS-CoV-2 infection (PMID: 33316181).
I recommend to integrate and discuss some parts of these citations also in the introduction. Comparison with other testings and specificity is essential.
I suggest also to see, cite and integrate this: Cochrane corner: rapid point-of-care antigen and molecular-based tests for the diagnosis of COVID-19 infection (PMID: 33294111).
Discussion: please give a clear message of this paper: which is the best testing method? What should clinicians do?
Round 2
Reviewer 1 Report
Thank you, authors, for making appropriate changes to the manuscript and for addressing this reviewer's concerns.
Reviewer 2 Report
The authors completed and edited all the requirements. Congratulations!